# SAS: Structured Activation Sparsification

**Yusuke Sekikwa, Shingo Yashima**
DENSO IT Lab. Inc., Tokyo, Japan
`yusuke.sekikawa, yashima.shingo@core.d-itlab.co.jp`

## Abstract

Wide networks usually yield better accuracy than their narrower counterpart at the expense of the massive `mult` cost. To break this tradeoff, we advocate a novel concept of *Structured Activation Sparsification*, dubbed SAS, which boosts accuracy without increasing computation by utilizing the projected sparsity in activation maps with a specific structure. Concretely, the projected sparse activation is allowed to have $N$ nonzero value among $M$ consecutive activations. Owing to the local structure in sparsity, the wide `matmul` between a dense weight and the sparse activation is executed as an equivalent narrow `matmul` between a dense weight and dense activation, which is compatible with NVIDIA's *Sparse Tensor Core* developed for the $N:M$ structured sparse weight. In extensive experiments, we demonstrate that increasing sparsity monotonically improves accuracy (up to 7% on CIFAR10) without increasing the `mult` count. Furthermore, we show that structured sparsification of *activation* scales better than that of *weight* given the same computational budget. `https://github.com/DensoITLab/sas_`

## 1 Introduction

Modern deep neural networks (DNNs) consist of numerous matrix multiplications between input activation and weight matrices, making it challenging for DNNs to deploy on resource-constrained edge devices due to the enormous `mult` count. One could reduce the computation by narrowing the width of the activation channel; however, it trades off accuracy.

Researchers explore the way to speed up DNN inference from various perspectives such as quantization (Esser et al., 2019), knowledge distillation (Hinton et al., 2015), designing efficient model architectures (Liu et al., 2022; Howard et al., 2017), and structured weight sparsification (Wang et al., 2019; Tan et al., 2020; Li et al., 2022; 2016; Wen et al., 2016). This paper explores the unexplored spectrum, *Structured Activation Sparsification*, positioned at the opposite end of the structured weight sparsification.

### Weight Sparsification

One could obtain an efficient model by removing unimportant weights from a wide, high-capacity model. It can be categorized into *unstructured* and *structured*.

***Unstructured weight sparsification*** prunes individual weights independently based on the criteria for the importance of the weight connection (e.g., by their magnitude) (Frankle & Carbin, 2018; Yang et al., 2019; Wortsman et al., 2019). Although it could achieve an extremely sparse network without sacrificing accuracy, the sparsity does not translate to the practical speedup on vector-processing architectures (e.g., GPU) because it has no structure on the sparsity pattern, making the parallel execution difficult (NVIDIA, 2020). In fact, even when a matrix is highly sparse (e.g., >95% zero), wall-clock times for the sparse `matmul` on GPU (`cuSPARSE`) is still slower than dense operation (`cuBLAS`) (Shi et al., 2020).

***Coarse-grained structured weight sparsification*** removes unimportant weights such that the sparsified operation can be vectorized, which includes block pruning (Wang et al., 2019; Chen et al., 2022), kernel shape sparsity (Tan et al., 2020), and channel/filter pruning (Li et al., 2022; 2016; Wen et al., 2016). The resultant network's theoretical `mult` count

directly translates to actual speedup on commodity vector-type hardware; however, it hurts model performance more severely than the unstructured fine-grained sparsity.

***Fine-grained structured (semi-structured) weight sparsification*** aims at the best of both worlds. The $N : M$ structured weights sparsification (SWS) utilized the local sparse structure in weight to realize the efficient parallel execution of sparse `matmul`. The structure indicates that the sparse weight has $N$ nonzero value for every continuous $M$ element (section 5). NVIDIA commercialized this idea in Sparse Tensor Core[1]. It incorporates this local structure by the vectorized local switch, which selects corresponding $N$ elements from $M$ consecutive activation, realizing the dot-product using only $N/M$ `mult` count. They demonstrate the actual speedup of SWS close to the theoretical gain. This real speedup on the wide-spread hardware using local sparsity patterns motivates us to explore the opposite: structured sparsity in activation, *expecting it to deliver better accuracy/speed tradeoff.* Hereafter, we call the *N:M* structured pattern as structured for brevity.

ACTIVATION SPARSIFICATION

Sparsity also exists in activation; however, unlike sparsity in weight, much less attention has been given to exploiting sparsity in activation for DNN speedup.

***Unstructured activation sparsification*** has been utilized to speed up the DNN inference. In many settings, DNNs' activation tends to be naturally sparse by the activation functions, e.g., rectified linear unit (`ReLU`). Most of the prior works utilized the activation sparsity by their novel hardware architecture (Zhu et al., 2022; Albericio et al., 2016; Rhu et al., 2018; Han et al., 2016; Parashar et al., 2017; Park et al., 2017; Georgiadis, 2019; Wang et al., 2021). (Kurtz et al., 2020) demonstrate the wall-clock speedup on a general-purpose CPU.

However, the sparsity induced by activation functions such as `ReLU` depends on its input; therefore, the resultant sparsity pattern is *unstructured*. This *unstructured* nature makes it difficult to parallelize the operation on vector-type processors such as GPU (Shi et al., 2020). This incompatibility with the popular vector-type processors partly explains why activation sparsity has been less explored than weight sparsity.

***Structured Activation Sparsification (SAS)*** is an unexplored area that we expect to pioneer the new frontier for accuracy/speed tradeoff in vector-processing architecture. Our contribution is two-fold: ①We introduce the idea of SAS, open up a new research area, and ②We invent a mechanism called structured sparse projection to construct SAS, realized vectorized `matmul` with sparse activation on GPU for the first time. Realizing structured sparsity, i.e., controlling the nonzero counts within consecutive activation elements, is difficult because the activation value depends on its input. We realized the structured sparsity in activation by *implicitly projecting* the input narrow/dense activation into higher dimensional space such that the projected activation has a local sparsity pattern. This contrasts to *pruning* as in SWS, which sparsify the value (weight) to be fixed after training. SAS yields better accuracy by utilizing the *increased flexibility* of the wide weight while keeping the same `mult` count. In other words, SAS *selects the appropriate weight depending on its input using the activation's sparsity pattern.* Importantly, owing to the local structure of the projected sparsity, SAS realized the vectorized `matmul` with sparse activation on typical GPU by utilizing the local routing mechanism originally developed for SWS.

Our experiments (section 3-4) demonstrate that increased sparsity in the SAS network enhances its capacity and accuracy monotonically without increasing `mult` count; furthermore, it performs much better than the SWS for the same speed.

## 2 STRUCTURED ACTIVATION SPARSIFICATION

A matrix multiplication (`matmul`) is a fundamental building block of neural networks. This study mainly focuses on the `matmul` that appears in convolution[2]. Convolution with the activation $\boldsymbol{I} \in \mathbb{R}^{C_i \times W \times H}$ and weight $\boldsymbol{W} \in \mathbb{R}^{C_o \times C_i \times k \times k}$ is equivalent to `matmul` between

---

[1] Ampere and beyond support this for 2:4 and 1:2 sparsity (NVIDIA, 2020).

[2] The same discussion can be applied to the `matmul` in most DNN elements, such as attention.

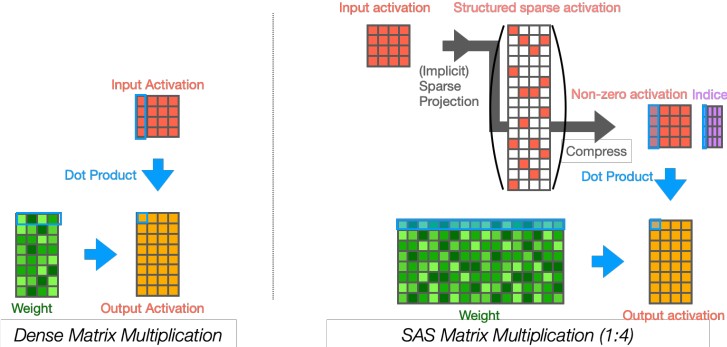

Figure 1: **SAS overview.** Conventional `matmul` between a dense/narrow weight and a dense/-narrow activation has a lower expressive power (left). The `matmul` between dense/wide weight and structurally sparse/wide activation composed by implicit sparse projection (SAS) has a higher expressive power while requiring the same `mult` count and comparable inference time (right).

unfolded activation $\boldsymbol{X} \in \mathbb{R}^{C_i kk \times WH}$ and the reshaped weight $\boldsymbol{W} \in \mathbb{R}^{C_o \times C_i kk}$:

$$\boldsymbol{O} = \boldsymbol{I} * \boldsymbol{W} = \boldsymbol{W}\boldsymbol{X}. \tag{1}$$

For dense weight $\boldsymbol{W}$ and dense activation $\boldsymbol{X}$, the `matmul` consumes $C_i kk \times C_o \times HW = \bar{C}_i \times C_o \times HW$ `mult` count. One can reduce the computation by narrowing the network width; `mult` count is reduced by a factor or $\alpha^2$ by narrowing the number of input and output channels by $\alpha$. Unfortunately, but apparently, this trades off accuracy.

## 2.1 Structured Sparsity in Activation

We study an unexplored area, exploiting structured sparsity in activation to realize a neural network model that is not only low FLOPS but also vectorizable. Concretely, the sparse activation is allowed to have $N$ nonzero value among $M$ consecutive activations. Owing to the sparsity, a `matmul` between sparse/wide activation $\tilde{\boldsymbol{X}}$ and dense/wide weight $\tilde{\boldsymbol{W}}$ could be reduced into the `matmul` between dense/narrow activation $\boldsymbol{X}$ and dense/narrow weight $\boldsymbol{W}$ as: $\boldsymbol{O} = \tilde{\boldsymbol{W}}\tilde{\boldsymbol{X}} = \boldsymbol{W}\boldsymbol{X}$. More importantly, it realizes the parallel vectorized execution owing to the local sparsity pattern, specifically $\boldsymbol{W}$ is constructed by locally routing the weight elements from the sparse weight $\tilde{\boldsymbol{W}}$ that correspond to the nonzero activation elements in $\tilde{\boldsymbol{X}}$. Actually, this operation is compatible with the commonly used GPU (section 2.3).

## 2.2 SAS by Sparse Projection

*How can we construct such structurally sparse activation?* Realizing structured sparsity (controlling the number of nonzero counts within consecutive activation elements) is challenging because, unlike the case of SWS, the activation changes depending on the input. Therefore, existing methods utilizing activation sparsity are *unstructured*, such as induced by `ReLU` (Shi et al., 2020), which is hard to vectorize on GPU. One could construct a structurally sparse activation by *max-pooling* along the blocked channel dimension; however, we can not expect this to achieve a good accuracy/speed tradeoff. Because ①we can not control the input-dependant activation to have one ($N$) significant value among $M$ consecutive element (otherwise, it removes crucial information); ②it discards most of the value in activation map (e.g., 94% when $M$=16) that is computed using valuable resources.

To realize the structured sparsity, we propose structured sparse projection $\mathcal{S}$ as follows:

$$\mathcal{S} : \boldsymbol{X} \in \mathbb{R}^{\bar{C}_i \times HW} \mapsto \tilde{\boldsymbol{X}} \in \mathbb{R}^{M/N \bar{C}_i \times HW}. \tag{2}$$

In the SAS projection $\mathcal{S}$, a single element in the source activation $\boldsymbol{X}$ is projected into one of an $M$ consecutive elements in target sparse activation $\tilde{\boldsymbol{X}}$ indexed by $\mathcal{I}$. In this study, we use the sign of $\log_2 M$ neighbor activation to compute the index $\mathcal{I}_{j,u}$ for $j$-th input channel and $u$-th spacial location as follows:

$$\mathcal{I}_{j,u} = \sum_{i=0}^{\log_2 M - 1} (\boldsymbol{X}_{j+i,u} > 0)\, 2^i. \tag{3}$$

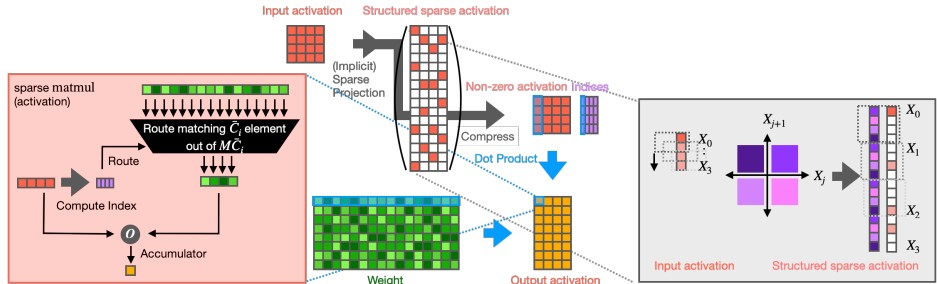

Figure 2: **SAS mechanism.** The `matmul` between sparse/wide activation $\tilde{X}$ and dense/wide weight $\tilde{W}$ could be vectorized on Sparse Tensor Core by selecting elements from the weight $\tilde{W}$ that correspond to the nonzero values in the activation $\tilde{X}$, skipping the unnecessary `mult` by zero (left). A dense/narrow activation $X \in \mathbb{R}^{\bar{C}_i \times HW}$ is projected to a structurally sparse/wide activation $\tilde{X} \in \mathbb{R}^{M\bar{C}_i \times HW}$ (here $M$=4). Local index $\mathcal{I}$ for projecting the activation is computed by using the sign of the $\log_2 M$ consecutive activation using eq. (3) (right). Note: we do not compute the sparse activation $\tilde{X}$ explicitly during inference; instead, $X$ and $\mathcal{I}$ is directly passed to the processor.

```
1   def sas_matmul(W, X, kSparse=2): # Activation X and Weight W, kSparse=2 indicates 50% sparsity
2       beta, gamma, alpha  = W.shape[0]*kSparse, W.shape[1], X.shape[1]
3       Y  = zeros([beta, alpha], dtype='float32')    # Output
4       E_ = zeros([gamma//kSparse//2, alpha], dtype='uint8') | 0x4
5       # Compute index by evaluating sgn−bit (eq. (3)) and Convert to 4−bit index
6       E_[signbit(X)] = 0xe
7       E = bitwise_or(E_[0::2,:], E_[1::2,:]<<4)
8       E_reodered = take(E, array(numpy.load('{}_{}.npy'.format(m, k)))) # need re−oder (supp. E, fig. A2)
9       # Copy compressed activation to SparseTensorCore
10      X_compressed = zeros((alpha*gamma//kSparse)*(4+1/2), dtype='uint8') # value followed by index
11      cuda.runtime.memcpy(X_compressed.data.ptr, X.data.ptr, (int)(alpha*gamma//kSparse)*4, ...)
12      cuda.runtime.memcpy(X_compressed.data.ptr+dat_size, E_reodered.data.ptr, (alpha*gamma//kSparse//2), ...)
13      # Execute matmul: Y = W @ X by calling cusparseLtbetaatmul
14      cusparselt.matmul(..., W.data.ptr, X_compressed.data.ptr, ..., Y.data.ptr, ...)
15      return Y
```

Listing 1: `cuSAS`: General SAS `matmul` library for Sparse Tensor Core. It is based on CuPy Okuta et al. (2017) wrapper of `cuSPARSELt`. The index needs to be reordered before execution of sparse `matmul` (L8). Refer to supp. E for more details about the reorder specific to NVIDIA GPUs.

The computational cost for the index is negligible (fig. 3 report its wall-clock time); it merely looks at the sign bit of the $\log_2 M$ neighbor activation (fig. 2). **Note**: the SAS using this projection includes the ReLU network as a particular case (e.g., when odd elements of weight are zero for $M$=2). Other than this simple indexing, there are various options; one could choose $N$ elements instead of 1 or incorporate another strategy for computing the index, e.g., estimating the index $\mathcal{I}$ using the previous layer's output (section 6.1).

**FLOPS.** The size of weight $\tilde{W}$ for the SAS network increases linearly with $M$; nevertheless, the number of `mult` count is the same as the base dense/narrow network, i.e., $\bar{C}_i \times C_o \times HW$. The `mult` count is also approximately the same with the SWS network (Zhou et al., 2021) with $\sqrt{M}$ times wider input/output channel for 1:$M$ sparsity pattern (Refer to supp. B for more detail). Both SAS and SWS require additional costs for routing the weight or activation. *Sparse Tensor Core* is highly engineered for this purpose and can execute the routing in parallel (section 2.3), and as reported in (NVIDIA, 2020), the time for the routing is small compared with the main `matmul`. The SAS network requires another negligible cost for computing index $\mathcal{I}$ of eq. (3) (fig. 3 reports the wall-clock time on actual hardware).

**Memory.** Although the `mult` count is constant for the activation sparsity $M$, the memory footprint of SAS increases linearly. This contrasts with the SWS, where storage for weight at inference time is constant irrespective of weight sparsity $M$. **Note**: the SWS network also requires $M$ times more memory during training. SAS and the SWS require additional storage for storing the index; however, it is small, requiring only $\log_2 M$-bit per element.

## 2.3 Hardware Implementation and Speed Benchmarking

The SAS `matmul` could be vectorized on an ordinary GPU having a local routing mechanism. For example, NVIDIA GPU equips this mechanism. During inference, we do not explicitly construct the sparse activation $\tilde{X}$; instead, we directly compute index $\mathcal{I}$ from dense activation $X$ ($\tilde{X}$ is computed implicitly). The $\mathcal{I}$ and $X$ is transferred to the core, then the

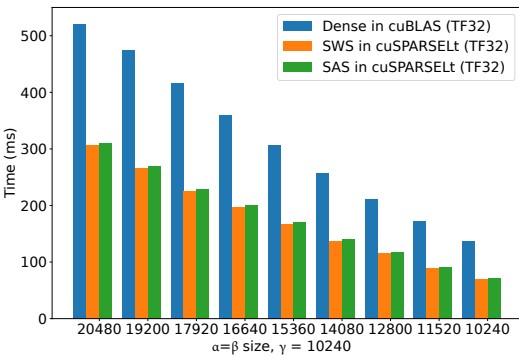

Figure 3: **Speed benchmarking.** SAS (1:2) vs SWS (1:2) for general `matmul` on NVIDIA A6000 GPU. Comparing the general `matmul` **WX** where $\mathbf{X} \in \mathbb{R}^{\gamma \times \alpha}$ and $\mathbf{W} \in \mathbb{R}^{\beta \times \gamma}$. The extra cost for SAS, c.f., index computation is less than 1.5% (when M = 20480) of the entire `matmul`. The time of cuBLAS is an estimation based on NVIDIA's white paper and included for reference.

vectorized switch routes the corresponding weight element using the local index, executing the sparse/wide `matmul` as dense/narrow `matmul`(fig. 2) in parallel. We developed a general `matmul` library for SAS (listing 1) called `cuSAS` for *Sparse Tensor Core*. `cuSAS` realized the vectorized `matmul` with structured sparse activation on GPU for the first time. Figure 3 report the wall-clock time of SAS for general `matmul` compared with SWS using the same matrix size. The difference between SWS and SAS comes from the overhead specific to SAS (index computation, reorder (supp. E), and its memory transfer), which is less than 1.5%, even with our naive implementation having several redundancies. ***Note***: the configuration of this experiment is different from the previous discussion and the other experiments using neural networks; we use the same matrix size for dense, SWS, and SAS to reveal their difference more clearly. Therefore, `mult` count of SWS and SAS is $1/M$ of that of dense when considering the sparsity. Refer to supp. C for more detail about the speed benchmarking.

### 2.4 TRAINING SAS NETWORK

One could transform most existing (narrow) neural networks into (wide) SAS networks to increase their expressive power without increasing inference speed. The conversion is simple: replacing the original non-linearity (e.g., `ReLU`) with our SAS projection $\mathcal{S}$. During the training, SAS `matmul` is computed as follows: it first projects the dense/narrow activation map into a structurally sparse/wide space by $\mathcal{S}$, constructing the sparse activation explicitly; then, it performs the conventional dense `matmul`. It is equivalent to the sparse `matmul` for efficient inference using hardware support (fig. 2).

**Optimizer for SAS**  To train the SAS network, one wants to utilize modern optimizers such as Adam (Kingma & Ba, 2014) and AdamW (Loshchilov & Hutter, 2019), which use adaptive learning rates based on the statistics of the gradient history. However, it is known that the adaptive learning rate could be unstable in the early stage of training, where the number of experienced gradients is small (Liu et al., 2020). This is more problematic in the SAS network (when $M$ is large) because each weight element receives gradients less frequently ($\times 1/M$ on average) than the dense network. To achieve stable training of the SAS network, we propose *Experience-RAdam* (ERAdam), which slightly modifies the formulation of RAdam (Liu et al., 2020). Basically, RAdam rectifies Adam(W)'s adaptive learning rates by multiplying $r_t < 1$ to reduce the variance in the early stage of training:

$$\rho_\infty = 2\,(1 - \beta_2) - 1, \ \rho_t = \rho_\infty - 2t\beta_2^t\left(1 - \beta_2^t\right), \tag{4}$$

$$r_t = \sqrt{((\rho_t - 4)\,(\rho_t - 2)\,\rho_\infty)/((\rho_\infty - 4)\,(\rho_\infty - 2)\,\rho_t)}, \tag{5}$$

where $t$ is the optimization steps, and $\beta$ is the hyperparameters used in computing the running average of gradients. As $t \to \infty$ (training proceeds), we have $r_t \to 1$, and it recovers the original Adam(W)'s update.

Our ERAdam scales steps $t$ for calculating $r_t$ of each weight element so that it is proportional to the number of received gradients. Specifically, we use the following scaled time vector

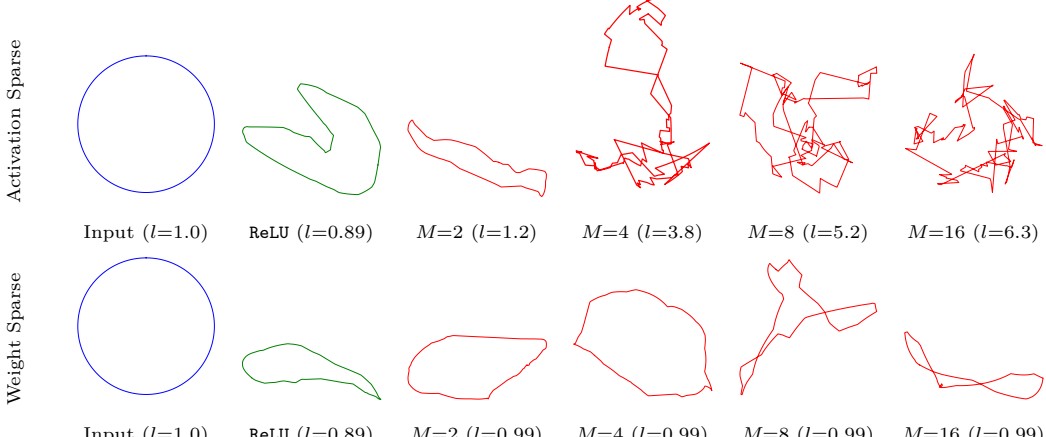

Figure 4: SAS capacity analysis by trajectory length. Comparing SAS (top) with dense `ReLU` network and SWS network (bottom). $l$ indicates the average relative length concerning the input circle length, where a longer length (complex trajectory) indicates more expressive power.

$\mathbf{t} \in \mathbb{R}^{\bar{C}_i}$ instead of $t$:

$$\mathbf{t}_i \leftarrow \mathbf{t}_i + \left( \sum_u (\nabla \tilde{\boldsymbol{X}}_{\cdot,u}) \neq 0 \right) / HW, \tag{6}$$

where $\nabla \tilde{\boldsymbol{X}} \in \bar{C}_i \times HW$ is gradient of activation corresponding to the weight $\boldsymbol{W}$. We find the optimizer with the CosineDecay (Loshchilov & Hutter, 2017) scheduler with k-decay (Zhang & Li, 2020) works well, especially for large $M$.

## 3  EXPRESSIVE POWER ANALYSIS

We propose SAS for DNNs, but it is difficult to discuss its superiority (or inferiority) over SWS based on their accuracy on a specific task; the result often turns back depending on the model structure, training strategy (scheduler, optimizer, etc), and dataset. To evaluate the difference of expressive power with SWS independently from these factors, we utilize the *Trajectory Length* (Raghu et al., 2017), which evaluates the expressiveness by measuring the length of the output trajectory given input sweeps along a one-dim path (e.g., circle).

We construct a two-layer neural network with 2-dim input and output (fig. A1). The base network has 32-dim intermediate channels, followed by the `ReLU` activation. We replace the `ReLU` by our SAS with different sparsity $M$, such that they all have the same `mult` count, i.e., structurally sparsified activation $\tilde{\boldsymbol{X}}$ has 32 nonzero elements. We also construct the SWS variant (Zhou et al., 2021), which has approximately the same `mult` count (supp. section A). We evaluate their trajectory length by inputting the one-dimensional circular path to the networks by varying the sparsity $M$ and report their average output trajectory length by changing the initial random weights.

Figure 4 report the results. The trajectory length of the SAS network increases with increasing sparsity $M$. On the other hand, the length from the SWS network is almost constant for $M$. This trajectory analysis suggests our SAS has more expressive power than the SWS, given the same computational budget at the expense of increased memory requirement.

## 4  EXPERIMENT

> *Given the same `matmul` budget, enlarging a model by SAS with sparse projection improves accuracy? Is it better than the SWS using the same hardware?*

We conducted extensive experiments to answer the questions. For the first question, we evaluated the accuracy by changing the sparsity $M$, keeping the same `mult` count. For the second question, we compared the SAS with SWS (Zhou et al., 2021), which has the same sparsity and `mult` count; both utilize the same hardware mechanism (i.e., local routing).

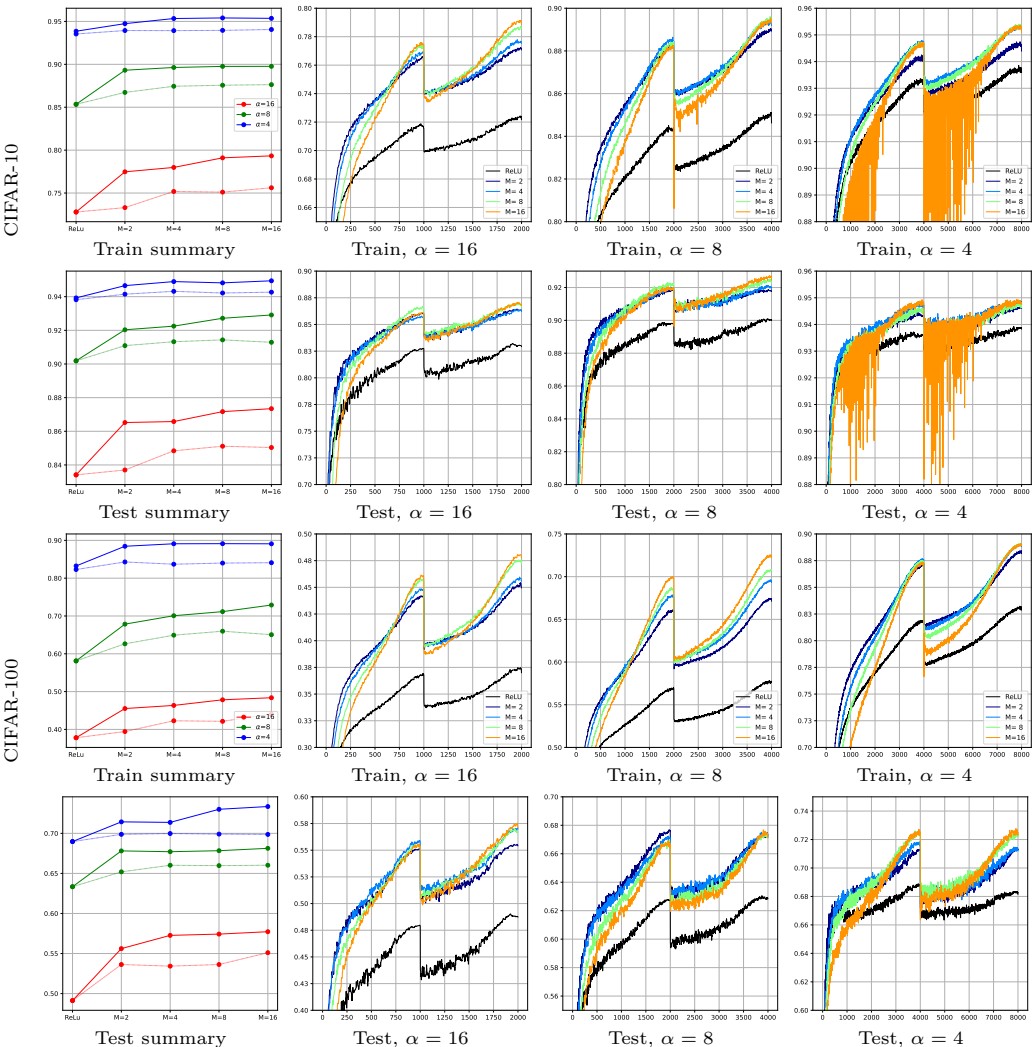

Figure 5: Accuracy of $\times\alpha$ narrow ResNet18 on CIFAR-10 and CIFAR-100 for difference SAS sparsity $M$. The summary plot reports the best score for each setting where dashed lines are the SWS network having the same sparsity in *weight*, which consumes the same `mult-add` count.

### 4.1 Experimental Setup and Results

We evaluate the classification accuracy on CIFAR-10/100 (Krizhevsky et al., a;b), and ImageNet (Deng et al., 2009). Because the commonly used network already achieves good saturated accuracy, we adopted the narrow version as a base network to better evaluate the effects of using the sparse activation or weight. Specifically, as a base dense `ReLU` network, we've utilized the network with $\alpha\times$ narrow width ($W \in \mathbb{R}^{C_o/\alpha \times \bar{C}_i/\alpha}$) except for the first and last layers; it approximately consume $\alpha^2\times$ less `mult` count. All variants, `ReLU`, SWS (Zhou et al., 2021) and SAS networks, have approximately the same `mult` count for the same $\alpha$. The 1:$M$ SWS network employs $\sqrt{M}\times$ more channel to make the `mult` count the same as the base `ReLU` network (section 2.2, supp. B).

For fair comparisons, we employ the same training strategy for all variants, i.e., the training epochs, batch size, optimizer, scheduler, etc. For training the SWS network, we employed the training method of (Zhou et al., 2021). We train the networks from scratch for all experiments. Refer to supp. D for more detail about the experimental configuration.

***CIFAR-10 / CIFAR-100***. We use ResNet18 (He et al., 2016) as one of the most popular architectures. For all the variants, we utilize our proposed ERAdam optimizer (section 2.4) in combination with k-decay scheduler (Zhang & Li, 2020); it is equivalent to RAdam for

the base network because $\sum_u (\nabla \tilde{\mathbf{X}}_{\cdot,u}) \neq 0$ in eq. (6) is approximately same for each weight elements. We trained them for $16 \times 1000/\alpha$ epochs, utilizing the data argumentation adopted in the ConvNeXt (Liu et al., 2022). We evaluate the accuracy by changing the base network width $\alpha$ and sparsity $M$. Figure 5 summarize the results.

***ImageNet***. We use ConvNeXt (Liu et al., 2022), which is computationally efficient by its network architecture design; e.g., it heavily utilizes depthwise convolutions to reduce the number of `mult`. For this ImageNet experiment, we conducted an experiment only on the practical configuration (rather than changing $\alpha$ and $M$ in wider rage); we use ConvNeXt-B, having half of the original input channel ($\alpha = 2$) as a base network and use $M = 2$ for SWS and SAS. We use their official code base only changing its activation and number of training epochs and keeping the rest of the highly optimized settings unchanged. The accuracy of the base network, SWS, and SAS is 80.7 %, 81.5 %, and **82.2** %, respectively.

## 4.2 DISCUSSION

As expected, the accuracy improves as we increase the activation's sparsity $M$. For all ranges of the base network width (having different $\alpha$), we observed the most significant gain from the base `ReLU` network to a $M{=}2$ sparse network. The improvement saturates around $M{=}8$ or $M{=}16$. Similarly, we observe a significant gain when the base network capacity is small (large $\alpha$). From a perspective of the training statistics evolution, the accuracy of the SAS network having significant sparsity (large $M$) is worse than the less sparse network (small $M$) at the beginning of the training; however, it surpasses them as training proceeds; sparser network requires more training epochs to reach their best accuracy. We hypothesize this is partly due to the sparse gradient when $M$ is large. In comparison with the weight sparsification, SWS network (Zhou et al., 2021) also shows improvement for the weight sparsity $M$. Yet, SAS scales significantly better than SWS for the same `mult` count.

In summary, we see the SAS realized by the projected sparsity boosts accuracy without increasing `mult` count; furthermore, the improvement is monotonic w.r.t. $M$ and is much better than SWS. The index $\mathcal{I}$ changes depending on its input, suggesting *the different weight elements are dynamically utilized depending on their input* (section 5). We hypothesize that the increased flexibility by the increases memory for weight is a primary source of SAS's superior accuracy boost over the SWS. The result also aligns with the increased capacity of SAS evaluated by the trajectory length analysis (section 3).

## 5 RELATED WORKS

This section discusses the relation to other approaches for efficient DNN inference on GPU.

**Unstructured weight sparsity.** It is hard to utilize the unstructured sparsity for speedup on GPU (Shi et al., 2020). For example, the wall-clock time for `matmul` between an 8000×8000 matrix with a sparsity of 90% and a dense matrix with the same size takes 780ms by `cuSPARSE`, while the corresponding dense algorithm by `cuBLAS` only requires 121ms. Though the sparse operation reduces the `mult` count by 90%, the dense `matmul` is 7× faster, suggesting the difficulty of utilizing unstructured sparsity in vector processors due to several overheads such as indexing.

**Dynamic kernel.** SAS dynamically selects weight elements based on activation. In this regard, it relates to a series of research exploring the input-dependent weight to increase the network expressive power (Chen et al., 2020; Jia et al., 2016)., which also relates to the attention mechanism (Vaswani et al., 2017). Our SAS is different both in its motivation and mechanism (e.g., SAS selects weight instead of composing it on the fly). Possible future work would be to use attention for computing the index instead of the strategy in section 2.2.

**Quantization.** Discretizing the weight and/or activation into a low-bit representation is commonly used to speed up the DNN inference (Yin et al., 2019; Esser et al., 2019; Li & Baillieul, 2004); low-precision `mult` is more computationally efficient and consumes less energy than the floating point counterpart. Furthremore, when we adopt a lower bit weight,

memory footprints and cost for transferring the weight decrease. The main drawback of SAS is the increased memory footprint for the weight matrix ($\times M$); we'll explore the combination with the lower-bit weight matrix to mitigate the issue.

**Low-rank factorization.** Factorization of `matmul` into a low-rank presentation is often utilized to reduce the number of `mult` count. The factorization can be integrated into network design as depthwise or 1x1 convolution; they are utilized in efficient network designs such as MobileNet (Howard et al., 2017) or ConvNeXt (which we evaluate). Recent research realized the optimization of low-rank structure (Idelbayev & Carreira-Perpinan, 2020). Ours are orthogonal to these low-rank approaches and could be combined.

## 6   CONCLUSION

We propose SAS, an unexplored but effective approach for efficient DNN inference on vector-type processors. The evaluation reveals that sparsifying activation by structured projection improves accuracy while maintaining the same `mult` count and inference speed. Our idea is compatible with NVIDIA's commercial GPU for $M=2$. It is not limited to the specific device but is a good match for a wide range of vector processes; we expect it to open the door for the new algorithm and hardware utilizing the structured sparsity in activation.

### 6.1   LIMITATIONS & FUTURE WORK

**Advanced projection algorithm.** In this study, we employed a straightforward strategy to compute the index $\mathcal{I}$ (section 2.2). We employed this method because it is computationally cheap, simply looking at the sign bit of the neighbor activation. It has some drawbacks for $M>2$ when the neighbor elements are close to zero. When the neighbor oscillates around zero, the sparse activation $\bar{X}$ undergoes discontinuous changes, which may negatively affect the accuracy; this phenomenon is more likely to occur when $M$ is large. From another perspective, our current implementation determines the index by the simple rule, which is not learned end-to-end. Learning to compute the index by incorporating extra channels or attention-like mechanisms (Vaswani et al., 2017) might improve the accuracy.

**Combination with SWS.** We used the switching circuit developed for the SWS to utilize the sparsity in activation for speedup. Exploring the combination of SAS and SWS would be an exciting direction. Unfortunately, the current GPU supports sparse `matmul` where one of the matrices is sparse. More efficient DNN could be realized on future GPU, which supports `matmul` where both inputs are structurally sparse.

**Library for DNN framework.** We developed the CuPy (Okuta et al., 2017) based library `cuSAS` (listing 1) for general `matmul` for 1:2 sparse pattern, which runs on actual Sparse Tensor Core hardware and realized wall clock speedup; however, integration to popular DNN framework has not yet been completed. As an important future work, we will develop SAS (depth-wise) convolution for Pytorch (Paszke et al., 2019), TensorFlow (Abadi et al., 2015), and JAX (Bradbury et al., 2018) by CUDA programming. We expect slightly better results (in terms of the percentage of SAS-specific overhead) than the result in fig. 3 will be achieved with the custom CUDA kernel because we can integrate SAS-specific online computation such as index computation, reorder (section E), and memory copy.

**Hardware consideration for significant sparsity.** Current NVIDIA's GPU support only 1:2 sparsity for `tf32` or `float` matrices[3], which restricts the SAS in higher sparsity on actual hardware. In principle, larger sparsity in SAS does not increase the `mult` count but may increase the switching circuit's area, energy, and storage cost. Considering the saturated gain by increasing the sparsity (section 4), the best sparsity can be defined; we left this consideration for future work. Furthermore, hardware support could make the index computation (section 2.2) more efficient. In addition to the efficient implementation on existing GPUs, we'll explore the novel vector-type processor design specific to the SAS.

---

[3]`https://docs.nvidia.com/cuda/cusparselt`. It supports a 2:4 sparsity for lower bit inputs.

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

# SAS: Structured Activation Sparsification
## Supplemental Material

## A    DETAIL IN TRAJECTORY LENGTH ANALYSIS

Network design for the *Trajectory Length* analysis (section 3) is illustrated in fig. A1. We evaluate the length by randomly initializing the weight 100 times and reporting their average. We also compose the SWS network having the same sparsity, which consumes approximately the same `mult` count. The SWS network uses `ReLU` activation.

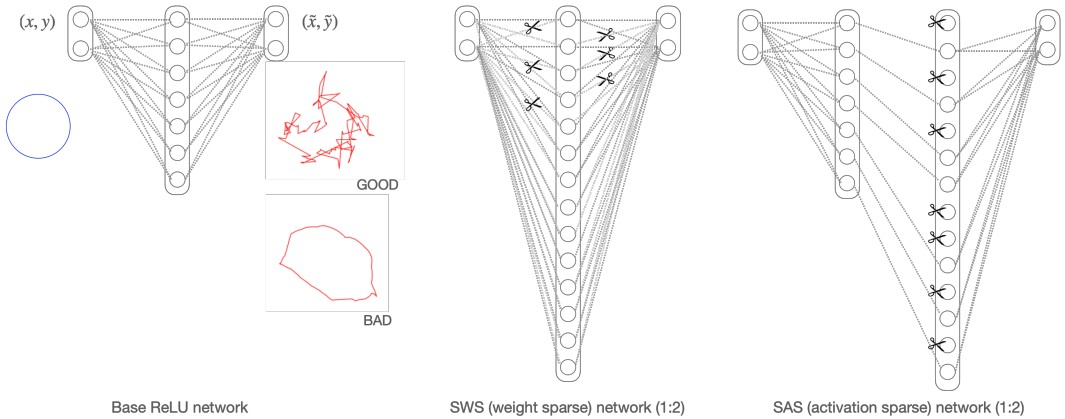

Figure A1: **Network design used for the *Trajectory Length Analysis*** (section 3). Relative output length with respect to the input circle length is an indicator of the network's expressive power; a longer length (complex trajectory) indicates more expressive power.

## B    DETAIL IN SWS NETWORK

In this paper, we propose SAS to improve the network capacity or accuracy without increasing the actual `mult` count. Therefore, we compare the SWS in the same scenario (keeping the `mult` count constant while increasing the sparsity). Specifically, we consider a base layer consisting of `matmul` between activation $\mathbf{X}$ and weight $\mathbf{W}$, where $\mathbf{X} \in \mathbb{R}^{\bar{C}_i \times HW}$ and $\mathbf{W} \in \mathbb{R}^{C_o \times \bar{C}_i}$ (eq. (1)). Using the proposed SAS, one could increase the network width for $M$ times while maintaining the same `mult` count as the base layer by utilizing the 1:$M$ sparsity pattern. The sparsified matrix shape is, $\mathbf{X} \in \mathbb{R}^{M\bar{C}_i \times HW}$ and $\mathbf{W} \in \mathbb{R}^{C_o \times M\bar{C}_i}$ (eq. (3)). Note that the SAS does not change the output channel dimension $C_o$.

On the other hand, in the case of SWS, if one increases the network width for the $M$ times and uses the 1:$M$ sparse pattern on weight, the `mult` count of the resultant network increases by about $M \times M/M = M$ because both input and output channel is $M$ times wider. In the case of SWS, by using $\sqrt{M}$ times wider network for the 1:$M$ sparsity pattern, we can construct the SWS network, which has roughly the same `mult` count as the base dense network and SAS. More specifically, we adopt the following configuration: We use $\sqrt{M}$ times (instead of $M$ times) wider input/output channel. In this configuration, we'll have approximately the same `mult` count as the base network. The network width needs to be an integer value, and it also needs to be a multiple of $M$. Hence, we use the weight having the shape of $\lfloor\sqrt{M}\bar{C}_i^{(l)}\rceil \times \lfloor\sqrt{M}C_o^{(l)}\rceil$ for the 1:$M$ SWS network, where $\lfloor\cdot\rceil$ is a rounding operator, $\bar{C}_i^{(l)}$, $C_o^{(l)}$ is the input/output channel dimension of $l$-th layer of the base dense network. For the last chunk, when it does not equal to $M$, we use $(\lfloor\sqrt{M}\bar{C}_i^{(l)}\rceil\lfloor\sqrt{M}C_o^{(l)}\rceil$

mod $M$):1 sparse pattern. This way, we construct the SWS network having approximately the same `mult` count as the base dense network.

For example, consider the $l$-th layer of the base network consisting of convolution with $\bar{C}_i$=288 ($C_i$=32, kernel size $k$=3) and $C_o$=128. When $M$=8, then we have $\lfloor\sqrt{M}\bar{C}_i^{(l)}\rceil$=815 and $\lfloor\sqrt{M}C_o^{(l)}\rceil$=181. Then `mult` count of the original dense layer (for single pixel) is 288×64=18432, the `mult` count of the weight sparse layer (SWS) is 18440 (815×181/8=18439 with modulo 3, we use $M$=3 for the last chunk).

## C  DETAIL IN SPEED BENCHMARKING

On the wall-clock speed benchmarking reported in section 2.3, we adopt the opposite configuration as the neural network experiment. We evaluate the speed by changing the sparsity pattern while keeping the matrix dimension unchanged to report more concrete comparisons.

Specifically, in the wall-clock speed benchmarking in fig. 3, we consider the `matmul` **WX** where $\mathbf{X} \in \mathbb{R}^{\gamma \times \alpha}$ and $\mathbf{W} \in \mathbb{R}^{\beta \times \gamma}$ **which is same for dense matmul, SAS matmul, and SWS matmul**. The `mult` count of the three variants is the same ($\gamma \times \alpha \times \beta$) when one does not take the sparsity into account. By utilizing the 1:2 fine-grained (or semi-structured) sparsity on *Sparse Tensor Core*, the `mult` count becomes half for SAS and SWS, i.e., $(\gamma \times \alpha \times \beta)/2$. As shown in fig. 3, we use fixed $\gamma$=10240, changing $\alpha = \beta$ from 10240 to 20480, which is the same for dense `matmul` SAS `matmul`, and SWS `matmul`.

We want to emphasize that the scenario in this speed benchmarking (keep the same matrix dimension) is different from the scenario for neural networks (keep the (almost) same `mult` count). We adopt the different configurations to evaluate the speed in a fair setting between SWS and SAS. In the neural network setting, one could construct the SWS network having approximately the same `mult` count with the base one by using $\sqrt{M}$ wider input/output channel (supp. B); however, it is hard to align the `mult` count precisely, and the shape of the matrices is different, which makes it hard to evaluate the overhead specific to SAS which we are interested in (computation of index, reorder, and memory transfer for the index). On the other hand, we can clearly evaluate this by measuring the time of the `matmul` of the same-sized matrix (section 2.3).

# D    Detail in Main Experiment

The primary experimental setup is summarized in table A1. The FLOPS and memory footprint of the network used in the experiments are summarized in table A2

For training the SWS network, we employed the method of (Zhou et al., 2021) instead of APEX's Automatic SParsity[1] because ①code base of (Zhou et al., 2021)[2] supports arbitrarily $1:M$ sparsity and ②it allows training from scratch which enables fair comparison with SAS.

Table A1: **Experimental setup**

|  | CIFAR-10 | CIFAR-100 | ImageNet |
|---|---|---|---|
| Network | | ResNet18 | ConvNeXt-B |
| Batch size | | 512 | 4096 |
| Training epochs | | $16/\alpha{\times}1000$ | 600 |
| Optimizer | | ERAdam (section 2.4) | AdamW |
| Scheduler | | Two cycle cosine with kDecay=2.0 Zhang & Li (2020) | Cosine |
| Initialization | | Kaiming-uniform He et al. (2015) | Truncated Gaussian |
| Base width $\alpha$ | | 4/8/16 | 2 |
| Sparsity $M$ | | ReLU/2/4/8/16 | ReLU/2 |

## CIFAR-10/CIFAR-100

The code for CIFAR-10 and CIFAR-100 is based on Pytorch lightning CIFAR10 tutorial code[3]. We use a single A6000 GPU for CIFAR10 and CIFAR100 experiments. It takes a day to train the single model.

## ImageNet

The code for ImageNet is based on ConvNeXt's (Liu et al., 2022) official code base[4]. We use four A100 GPUs (each holding 256 batches) with an update frequency of four to virtually construct the batch size of 4096. It takes about two weeks to train a single SAS model for 600 epochs (double the original 300 epochs to compensate for sparse gradient); we use the default setup of ConvNeXt's (Liu et al., 2022) official repository for training ImageNet-1K (without pre-training using ImageNet-22K), only changing the original dense `matmul` to our proposed SAS `matmul`(conv2D → SASconv2D, and linear → SASlinear) and training epochs $(300 \rightarrow 600)$. We use the original AdamW optimizer to keep the original ConvNeXt's highly optimized settings intact as much as possible (furthermore, in this moderate sparsity of $M$=2, proposed ERAdam behaves almost identical to AdamW with warm-up). Refer to their paper for a more detailed setup.

Table A2: **FLOPS and memory footprints**. Note that FLOPS and the number of parameters and FLOPS of SWS are not precisely the same as the base dense network as discussed in supp. B, but the difference is less than 1% for all the configurations.

| Network | FLOPS (all) | Params (dense, SWS) | Params (SAS) |
|---|---|---|---|
| ResNet18 ($\alpha$=4) | 114K | 731K | 731K$\times M$ |
| ResNet18 ($\alpha$=8) | 28K | 182K | 182K$\times M$ |
| ResNet18 ($\alpha$=16) | 7K | 46K | 46K$\times M$ |
| ConvNeXt-B ($\alpha$=2) | 3850M | 22M | 22M$\times M$ |

---

[1] http://github.com/NVIDIA/apex

[2] http://github.com/NM-sparsity/NM-sparsity

[3] http://lightning.ai/docs/pytorch/stable/notebooks/lightning_examples

[4] https://github.com/facebookresearch/ConvNeXt

TRAINING

During the training, SAS `matmul` is computed as follows: it first projects the dense/narrow activation map into a structurally sparse/wide space by $\mathcal{S}$, constructing the sparse activation explicitly; then, it performs the conventional dense `matmul`. Note that it is equivalent to the sparse `matmul` for efficient inference when hardware support is available, and we do not construct the sparse matrix explicitly during inference. Still, the hardware directly processes the narrow dense feature along with the index computed online (fig. 2). Refer to the pseudo-code in listing 2 during training. When implemented this way, the gradient for the wide weight $\tilde{\boldsymbol{W}}$ could be computed using *autograd* mechanism.

```python
1  def SAS_proj(x, m): # m corresponds to log2(M) in the main text
2      B, C, H, W = x.shape
3      xa = [torch.roll(x[:, None], i, dims=1) for i in range(m)]
4      ind = torch.cat([2**i*(torch.signbit(x_)) for i, x_ in enumerate(xa)], dim=2).sum(dim=2, keepdim=True)
5      x_sparse = torch.zeros([B, 2^m, C, H, W]).scatter_(1, ind, x[:, None]).view([B, (2^m)*C, H, W])
6      return x_sparse
7  def forward(self, x): # x: input activation, m: sparsity factor (actual sparsity is 100(1−1/2^m)[%])
8      return F.conv2d(SAS_proj(x, m), self.weight, bias=self.bias, stride=....)
```

Listing 2: Code of SAS `conv2d` layer for training (PyTorch). Note: During inference, one does not need to construct the sparse activation explicitly (L5); refer to fig. 2 for efficient infrence mechanism.

## E    MEMORY ARRANGEMENT FOR cusparseLtMatmul

The *Sparse Tensor Core* and the `cuSPARSELt` library were originally developed to speed up the DNN having structured weight sparsity (SWS). Our SAS improved the accuracy/computation tradeoff by using **the same hardware and software library**. In the case of SWS, the index could be precomputed after training using `cusparseLtSpMMACompress` function of `cuSPARSELt` library. Figure A2 illustrates the memory arrangement of value and index for `cusparseLtMatmul` operation for NVIDIA's *Sparse Tensor Core*. The value matrix follows the index matrix. In the case of a 1:2 sparse pattern (TF32), the index is stored using 4-bit (although it can be represented in 1-bit; *Sparse Tensor Core* use 4-bit to index a 1:2 pattern), and '0x4' and '0xe' is assigned for indexing the first and second element, respectively.

The important note for *Sparse Tensor Core* is that the order of the index is not aligned with its corresponding activation value; it needs to be reordered (L8 in listing 1), and the reordered index $\tilde{\mathcal{I}}$ needs to be supplied to the core to get the correct result. The arrangement depends on the size of activation matrices X (The memory arrangement in fig. A2 illustrates the specific case when the input matrix is $64 \times 32$).

### HOW TO USE THE cusparseLtMatmul FOR SAS?

One can also use the `cusparseLtSpMMACompress` function for SWS to executed the SAS `matmul`, ①compute the index $\mathcal{I}$ using equation 3, then ②explicitly computing the sparse activation $\tilde{X}$ and finally ③compute the reordered index $\tilde{\mathcal{I}}$ using `cusparseLtSpMMACompress`. However, it is redundant and inefficient. We already have compressed activation $X$ as an output from the previous layer; the index $\mathcal{I}$ is computed cheaply. We want to reorder the index $\mathcal{I}$ to get $\tilde{\mathcal{I}}$ without explicit construction of sparse activation $\tilde{X}$ as we discussed in the main text (section 2.3). The problem is that NVIDIA does not provide information about the rendering mechanism of `cuSPARSELt`.

### TOOLS FOR DIGGING UP THE REORDERING MATRIX

We developed a helper tool of `cuSAS`[5] for finding out the rendering matrix $O$ for arbitrary size matrix to realize a general `matmul` for SAS activation. The core idea for realizing the elucidation is *impulse response* of `cusparseLtSpMMACompress` function. Specifically, we input the sparse activation $\tilde{X}$; all the odd element has a non-zero value except the $(i, j)$ element. Then, looking at the reordered index computed by `cusparseLtSpMMACompress`, we can find the destination index $(\tilde{i}, \tilde{j})$ where $(i, j)$ element in the original index should be warped. Repeating this process for all the row-column pairs, we get the reordering matrix $O$ such that $\tilde{\mathcal{I}} = \mathcal{I}[O]$. The reordering matrix $O$ depends only on the size of $X$; therefore could be precomputed. We'll also open-source this tool.

It is possible to implement the CUDA kernel, which runs the following operation at once for more efficient SAS `matmul`; ①checks the sign bit, ②assign either '0x4' or '0xe', and ③reorder.

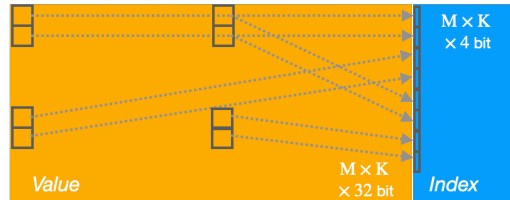

Figure A2: **Memory arrangement for cusparseLtMatmul.** The index is located just after the value. The index needs to be arranged for the execution of sparse `matmul` using `cusparseLtMatmul`. The formatting of the index depends on the size of the matrix X in listing 1. This figure illustrate the case for the 1:2 sparse pattern (TF32) and the input matrix size is $32 \times 64$.

---

[5]The current version supports sparse activation with a 1:2 structured pattern.

# F Fine-grained (semi) Structured Weight Sparsity.

NVIDIA's SWS (NVIDIA, 2020) could speed up the `matmul` with weight having moderate rate sparsity (e.g., 50%) on GPU, which is almost impossible for the unstructured sparse pattern (section 5). They realized actual speed up by utilizing a specific pattern in their sparsity, namely $N : M$ structured sparsity. Suppose a typical matrix multiplication between activation $\boldsymbol{X} \in \mathbb{R}^{16 \times 32}$ and weight $\boldsymbol{W} \in \mathbb{R}^{32 \times 8}$. The *Dense Tensor Cores* implement this `matmul` by two cycles. In contrast, the Sparse Tensor Core only needs one cycle if the weight tensor $\boldsymbol{W}$ satisfies the structured sparse pattern (fig. A3).

Our SAS could utilize the same hardware by the novel structured sparse projection mechanism. With the same computational budget and on the same hardware, SAS realizes better accuracy if one can use extra memory for storing the weight.

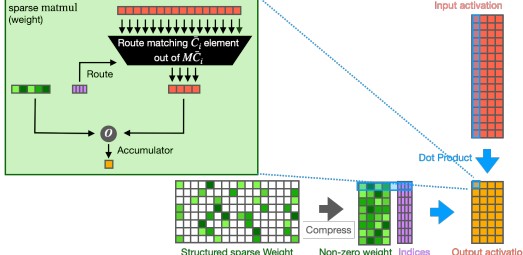

Figure A3: **SWS `matmul` on Sparse Tensor Core** NVIDIA (2020). Compare with our SAS `matmul` mechanism in fig. 2.

