# OpenReview forum: "SAS: Structured Activation Sparsification"
_ICLR.cc/2024/Conference — ICLR 2024 poster_

### Official Review · Reviewer_N3vE · 2023-10-30

**Soundness:** 2 fair
**Presentation:** 2 fair
**Contribution:** 3 good
**Rating:** 6
**Confidence:** 3

**Summary:**

This paper proposes a new concept of sparsity that maps the input activation values to a sparse representation, then exploits Nvidia Ampere sparsity for sparse computation by widening the weights. By doing this, wider weights make the network have stronger representation ability, and computation remains consistent because of the n:m sparsity operation. The authors conducted experiments on CIFAR and ImageNet datasets to verify the effectiveness of the proposed method.

**Strengths:**

1. The presented idea is clever and novel. It can effectively enhance the representation ability of the network under the same amount of computation compared with the vanilla network. I think this brings a nice insight to the field.
2. The author developed a general matmul library for the proposed SAS using Sparse Tensor Core. This gives the method practical value in the field, which is appreciated.

**Weaknesses:**

1. It is inaccurate for the author to state that the computation of SWS and SAS are the same. I can agree that the computation of SAS and the original dense network are the same, but the computation of SWS is obviously higher than that of SAS.
2. Let's continue considering this point. Figure 3 raises a big question for me. The author doesn't express the dimensions of the weights corresponding to SWS and SAS. Although it intuitively looks like SAS has a clear acceleration in the graph, I think this is unreasonable because the author also says the number of mult count in SAS is the same as the base dense/narrow network. This greatly reduces my enthusiasm for this paper.
3. The current presentation pf experiments is very scattered. I think the author should provide a comparison of the full network’s computation, inference speed, and accuracy to give readers a clearer comparison. For instance, when compressing ConvNeXt-b, although the accuracy of SAS is higher than SWS, a comparison of SAS's operation count and inference time also needs to be provided.

**Questions:**

Please see the weakness part.

---

> ### Author Response · Authors · 2023-11-17
> **Response (1/3)**
>
> We really appreciate the insightful comments by *N3vE*.
> Especially, *N3vE*'s comments helped us to clarify the potential inaccuracy and confusion about the experimental configuration.
> We consider that the following first three concerns kindly raised by *N3vE* come from essentially the same source of inaccuracies in the original manuscript regarding the comparison configuration of dense, SAS, and SWS: compare accuracy by increasing network width without increasing $\texttt{mult}$  count or comparing wall-clock time by changing sparse pattern while keeping the same matrix size.
> We revised the manuscript as follows, and we are now confident that the revised manuscript and the following answer clarify the concerns.
>
> >It is inaccurate for the author to state that the computation of SWS and SAS are the same. I can agree that the computation of SAS and the original dense network are the same, but the computation of SWS is obviously higher than that of SAS.
>
> >The author doesn't express the dimensions of the weights corresponding to SWS and SAS.
>
> >Although it intuitively looks like SAS has a clear acceleration in the graph, I think this is unreasonable because the author also says the number of $\texttt{mult}$ count in SAS is the same as the base dense/narrow network.
>
> In this paper, we propose SAS to increase the network capacity to improve the accuracy without increasing the actual $\texttt{mult}$ count.
> Therefore, we compare the SWS in the same scenario (keeping the $\texttt{mult}$ count constant while increasing the sparsity).
> Specifically, we consider a base layer consisting of $\texttt{matmul}$ between activation $\mathbf{X}$ and weight $\mathbf{W}$, where $\mathbf{X}\in \mathbb{R}^{\bar{C}_i\times {HW}}$ and $\mathbf{W}\in \mathbb{R}^{ {C}_o \times\bar{C}_i}$ (eq. (1)).
> As *N3vE* explains, using the proposed SAS could increase the network width for $M$ times while maintaining the same $\texttt{mult}$ count as the base layer by utilizing the 1:$M$ sparsity pattern.
> The sparsified matrix shape is, $\mathbf{X}\in \mathbb{R}^{M\bar{C}_i\times {HW}}$ and $\mathbf{W}\in {C}_o\times \mathbb{R}^{M\bar{C}_i}$ (eq. (2)).
> SAS does not change the output channel dimension ${C}_o$.
>
> On the other hand, in the case of SWS, if one increases the network width for the $M$ times and uses the 1:$M$ sparse pattern on weight, the $\texttt{mult}$ count of the resultant network increases by about $M\times M/M=M$ because both input and output channel is $M$ times wider.
> We consider *N3vE*’ kindly point out this issue.
>
> In the case of SWS, by using $\sqrt{M}$ times wider network for the 1:$M$ sparsity pattern, we can construct the SWS network, which has roughly the same $\texttt{mult}$ count as the base dense network and SAS network (section 2.3).
> Thanks to the questions raised by *N3vE*, we find the original manuscript was unclear from three perspectives: 1) the explanation in the original manuscript "$\sqrt{M}$ times wider input/output channel" was insufficient and needs more detail, 2) Although we adopt the opposite benchmarking configuration for the speed benchmarking in fig. 3 (we keep the same matrix dimension while changing the sparse pattern), we failed to explain this difference clearly, which may confuse the reader about the interpretation of the speed benchmarking result (and also the accuracy benchmarking result), 3) The reason why we evaluate the speed using the  $\texttt{matmul}$  of the same shaped matrix was not clearly stated.
> We've updated the manuscript to clarify the above three issues as follows;
>
> ###  Issue 1) Configuration of SWS network
> The original manuscript lacks the following detailed discussion.
> When the network is $\sqrt{M}$ times (instead of $M$ times) wider with 1:$M$ sparsity pattern in weight, the $\texttt{mult}$ count will be the same as the base network.
> But, the network width needs to be an integer value, and it also needs to be multiple of $M$.
> Therefore, we use the weight having the shape of  $\lfloor\sqrt{M}\bar{C}_i^{(l)} \rceil\times \lfloor\sqrt{M}{C}_o^{(l)} \rceil$ for the 1:$M$ SWS network to get approximately the same $\texttt{mult}$ count as the base network, where $\lfloor\cdot\rceil$ is a rounding operator,  $\bar{C}_i^{(l)}$, ${C}_o^{(l)}$ is the input/output channel dimension of the base dense network.
> For the last chunk, when it does not equal to $M$, we use $(\lfloor\sqrt{M}\bar{C}_i^{(l)} \rceil  \lfloor\sqrt{M}{C}_o^{(l)} \rceil \mod M)$:1 sparse pattern.
> In this way, we construct the SWS network having almost the same $\texttt{mult}$ count as possible.

---

> ### Author Response · Authors · 2023-11-17
> **Response (2/3)**
>
> Let's consider a specific example,
> the $l$-th layer of the base network consisting of convolution with $\bar{C}_i$ = 288 ($C_i$ is 32, kernel size $k$ is 3) and $C_o$ = 64.
> When $M$=8, then  We have $\lfloor\sqrt{M}\bar{C}_i^{(l)} \rceil$=815 and $\lfloor\sqrt{M}{C}_o^{(l)} \rceil$=181.
> Then $\texttt{mult}$ count of the original dense layer (for single pixel) is 288$\times$64=18432, the $\texttt{mult}$ count of the weight sparse layer (SWS) is 18440 (815$\times$181/8=18439 with modulo 3, we use M=3 for the last chunk).
> The comparison of the layer shape in this example case is summarized as follows:
>
> |    |  input $\bar{C}_i\rightarrow \tilde{C}_i$ |  output ${C}_o$ |  FLOPS | weight shape  | activation shape  |\\\
> | -------- | ------- |------- |------- |------- |------- |------- |
> | base (dense)    | 288 | 64  | 18432  | 64$\times$288 |  288$\times$HW |
>  |SAS ($M$=8)   | 288 $\rightarrow$ 2304 | 64  | 18432 | 64$\times$2304  |  2304$\times$HW (1:8 sparse)|
> | SWS ($M$=8)   | 815 | 181  | 18440 | 181$\times$815 (1:8 sparse) |  815$\times$HW|
>
> ### Issue 2 Configuration for speed benchmarking
> On the wall-clock speed benchmarking reported in fig. 3, we use the opposite configuration, changing the sparsity pattern while keeping the matrix dimension unchanged to report more concrete comparisons.
> The original manuscript did not emphasize this significant configuration difference, leading to a misunderstanding about the interpretation of the results.
> Furthermore, symbols used to describe the matrix shape ($\mathrm{M}, \mathrm{N}$) were confusing with the ones used for specifying the sparsity pattern ($N, M$).
>
> In the wall-clock speed benchmarking in fig. 3, we consider the $\texttt{mult}$  $\mathbf{W}\mathbf{X}$ where $\mathbf{X}\in \mathbb{R}^{\mathrm{\gamma} \times \mathrm{\alpha}}$ and $\mathbf{W}\in \mathbb{R}^{\mathrm{\beta}\times \mathrm{\gamma}}$  **which is identical for dense $\texttt{mult}$, SAS $\texttt{mult}$, and SWS $\texttt{mult}$**.
> The $\texttt{mult}$ count of the three variants is the same $(\mathrm{\gamma}\times \mathrm{\alpha}\times \mathrm{\beta})$ when one does not take the sparsity into account.
> By utilizing the 1:2 fine-grained (semi-structured) sparsity on *Sparse Tensor Core*, the $\texttt{mult}$count becomes half for SAS and SWS, i.e., $(\mathrm{\gamma}\times \mathrm{\alpha}\times \mathrm{\beta})/2$.
> We show the number of dimensions of the weight and activation for wall-clock time benchmarking in the graph (we use fixed $\mathrm{\gamma}$=10240, changing $\mathrm{\alpha}=\mathrm{\beta}$ from 10240 to 20480), which is the same for dense $\texttt{matmul}$ SAS $\texttt{matmul}$, and SWS $\texttt{matmul}$.
>
> Again, we want to emphasize that the scenario in this speed benchmarking (keep the same matrix dimension) is different from the scenario for neural network (keep the (almost) same $\texttt{mult}$ count).
> The statement "the number of $\texttt{mult}$ count in SAS is the same as the base dense/narrow network" is not meant for this speed evaluation experiment but for our main application scenario of the SAS for neural networks (section 3, section 4).
>
> ### Issue 3 Reason for the speed benchmarking configuration
> As discussed above, we adopt the different configurations to evaluate the speed in a fair setting between SWS and SAS.
> We can construct an SWS network having approximately the same  $\texttt{mult}$ count with base dense network and SAS network by adjusting the network width (as discussed in issue 1); however, it is not precisely the same, and the shape of the matrices will be different (please refer to the table in issue-1), which makes it hard to evaluate the overhead specific to SAS which we are interested in (index related operation).
> While we can precisely evaluate this by measuring the time of the $\operatorname{matmul}$ of the same-sized matrix (section 2.3, supp. B).
>
> To clarify the above point, we've made the following changes.
> * Added the details about constructing the SWS network with the similar $\texttt{mult}$ count as discussed above (section 2.3). We've also clarified in the experimental section (section 3, section 4) that the $\texttt{mult}$ count of the SWS network is not exactly the same as the dense or SAS network, but it is approximately the same.
> * Moved the speed benchmarking to a separate paragraph and clarified the distinction between the two benchmark setups as discussed above (section 2.3).
> * Changed the symbols  of the matrix size from $\mathrm{K}, \mathrm{M}, \mathrm{N}$ to $\mathrm{\gamma}, \mathrm{\alpha}, \mathrm{\beta}$ to avoid the notational clutter with the sparsity pattern $M, N$ (section 2.3, fig. 3).

---

> ### Author Response · Authors · 2023-11-17
> **Response (3/3)**
>
> > I think the author should provide a comparison of the entire network’s computation, inference speed, and accuracy to give readers a clearer comparison. For instance, when compressing ConvNeXt-b, although the accuracy of SAS is higher than SWS, a comparison of SAS's operation count and inference time also needs to be provided.
>
> Relating to the above discussion, the $\texttt{mult}$ count for the SAS and SWS networks are (almost) the same for the neural networks experiment.
> In the case of ConvNeXt-b, the FLOPS of SAS and base dense network is 3.85M, and the FLOPS of SWS is also 3.85M.
> We've added the $\texttt{mult}$ count and number of parameters in the supp. D (tab. A2), including the ResNet18.
>
> We report the wall-clock speed benchmarking in fig. 3, demonstrating that SAS-based multiplication can speed up the $\texttt{matmul}$  on actual hardware.
> The overhead of SAS over SWS is the online computation of the index (eq. 3), reorder (supp. E), and the memory transfer of the index.
> Our experiment shows that it is only a faction (1.5\%) of the entire  $\texttt{matmul}$ in terms of the wall-clock time.
> However, as *N3vE* points out, we have yet to benchmark the speed as a whole neural network.
> There are two reasons why we evaluate the wall-clock time using  $\texttt{matmul}$ of the same-sized matrix (fig. 3) instead of the entire network.
>
> * It is hard to exactly match the $\operatorname{mult}$ count between SWS and SAS network, and the matrix shape vastly differs between the two, making the evaluation of SAS-specific computation (online index computation) time difficult (please also refer to the discussion in the previous question). In contrast, we can precisely evaluate this by measuring the $\texttt{matmul}$ time of the same-sized matrix (section 2.3, supp. B).
>  * Due to the limited high-level API to utilize the *SparseTensorCore* (e.g., *cuSPARSELt*. We need to develop a custom CUDA kernel using a low-level library (e.g., [Cutlass](https://github.com/NVIDIA/cutlass) to deal with the specific $\texttt{mult}$ operation such as convolution or depth-wise convolution to evaluate the wall clock time as a whole network.
>
> We are now developing a custom CUDA kernel and will publish the source code.
> We expect slightly better results (regarding the percentage of SAS-specific overhead) than the result in fig. 3 will be achieved with the custom CUDA kernel (section 2.3, section 6.1) because we can integrate index computation, reorder (supp. E), and memory copy, eliminating unnecessary DRAM access.
>
> Thanks to *N3vE*, we noticed the above-discussed critical discussion about the wall-clock benchmarking is insufficient in the original manuscript.
> We've added the discussion in section 2.3.

---

> > ### Comment · Reviewer_N3vE · 2023-11-21
> > **Official Comment by Reviewer N3vE**
> >
> > Thanks for the author's response, which has successfully laid to rest my concerns. Consequently, I am increasing my rating. Nonetheless, I enthusiastically encourage the author to refine their current method and experimental description, as it indeed could easily lead to confusion, akin to what I previously experienced.

---

> > > ### Author Response · Authors · 2023-11-21
> > > **Thank you for your response**
> > >
> > > Thank you for your kind response, and we are glad that the response clarified the concerns.
> > >
> > > >I enthusiastically encourage the author to refine their current method and experimental description, as it indeed could easily lead to confusion akin to what I previously experienced.
> > >
> > > We promise to further update the manuscript in method (section 2) and experiment (section 3, section 4) until the very end of the rebuttal deadline.
> > >
> > > Thank you again for helping us to improve that manuscript by pointing out the crucial issues that led to the potential misunderstanding (configuration for comparing SAS with SWS).

---

### Official Review · Reviewer_W2qU · 2023-10-30

**Soundness:** 3 good
**Presentation:** 3 good
**Contribution:** 2 fair
**Rating:** 6
**Confidence:** 4

**Summary:**

This paper proposed SAS, a method to explore structured sparse activation in CNN. SAS describes the approach to generate structured sparse activation, software and hardware implementations. Numerical experiments on image classification validates the efficacy regarding accuracy aspect.

**Strengths:**

- The paper is written well, and technically sound.
- The study problem is interesting and may deliver real impact onto DNN speedup.

**Weaknesses:**

- The terminology of structured sparsity is misleading. We typically refer the N:M sparsity as "semi structured sparsity" to distinguish it from standard structured sparsity including disjoint group sparsity, overlapping group sparsity and hierarchical sparsity.

- The realistic benefits of structured sparse activation is not clear. Although the topic is interesting, I am not sure what is the actual speedup gain of such sparse activation that can deliver to the community. The paper seems equipping without numerical results regarding speedup as well.

- The citation format is wrong. Please use \citep{} rather than \cite{} to cite references.

**Questions:**

- What is the training cost to yield SAS network compared to training as standard?

---

> ### Author Response · Authors · 2023-11-17
> **Response (1/1)**
>
> Thanks for the insightful and sharp comments.
> In particular, suggestions about the terminology of $N$:$M$ sparsity helped us a lot to clarify and avoid unintended confusion about the scope and the contribution of the proposed study.
> We are now sure that the revised manuscript incorporates the raised concerns.
>
> >The terminology of structured sparsity is misleading. We typically refer to the $N$:$M$ sparsity as "semi-structured sparsity" to distinguish it from standard structured sparsity including disjoint group sparsity, overlapping group sparsity and hierarchical sparsity.
>
> Thank you for the suggestion.
> We agree with *W2qU* that the term "semi-structured" well describes the $N$:$M$ sparsity.
> We used the term "Fine-grained structured" sparsity following [NVidia's official whitepaper](https://images.nvidia.com/aem-dam/en-zz/Solutions/data-center/nvidia-ampere-architecture-whitepaper.pdf) and use structured afterward for brevity (section 1).
> Thanks to the *W2qU*'s comments, we find the original manuscript was unclear on this point.
> We clarified the terminology in section 1 and added the note that this is also called "semi-structured" to give a clear distinction from the standard structured sparsity.
>
> >The realistic benefits of structured sparse activation is not clear. Although the topic is interesting, I am not sure what is the actual speedup gain of such sparse activation that can deliver to the community. The paper seems equipping without numerical results regarding speedup as well.
>
> We report the wall-clock speed benchmarking in fig. 3, demonstrating that SAS-based multiplication can speed up the $\texttt{matmul}$  on actual hardware.
> The overhead of SAS over SWS is the online computation of the index (eq. 3), reorder (supp. E), and the memory transfer of the index.
> Our experiment shows that it is only a faction (1.5\%) of the entire  $\texttt{matmul}$ in terms of the wall-clock time.
> However, we've evaluated the time for a single  $\texttt{matmul}$ (with varying sizes), and we have not benchmarked the speed as a whole neural network.
> There are two reasons why we evaluate the wall-clock time using  $\texttt{matmul}$ of the same-sized matrix instead of the entire network.
> 1.  It is hard to exactly match the $\operatorname{mult}$ count between SWS and SAS network, and the matrix shape differs largely between the two, making the evaluation of SAS-specific computation (online index computation) time difficult. In contrast, we can clearly evaluate this by measuring the time of the $\operatorname{matmul}$ of the same-sized matrix (section 2.3, supp. B). This discussion also relates to the *N3vE*'s question; please refer to the response for the specific example of why this is the case.
> 1. Due to the limited high-level API to utilize the *SparseTensorCore* (e.g., *cuSPARSELt*). We need to develop a custom CUDA kernel using a low-level library (e.g., [Cutlass](https://github.com/NVIDIA/cutlass) to deal with the specific $\texttt{mult}$ operation such as convolution or depth-wise convolution to evaluate the wall clock time as a whole network.
>
> We are now working on the development of the custom CUDA kernel and will publish the source code.
> We expect slightly better results (in terms of the percentage of SAS-specific overhead) than the result in fig. 3 will be achieved with the custom CUDA kernel (section 2.3, section 6.1) because we can integrate index computation, reorder, and memory copy.
>
> Thanks to the *W2qU*'s comment, we find that the original manuscript failed to discuss this important motivation about the wall-clock benchmarking.
> We've added the discussion in section 2.3 and section 6.1.
>
> >The citation format is wrong. Please use "citep" rather than "cite" to cite references.
>
> Thank you for pointing out this.
> We've fixed all the citation formats.

---

> > ### Comment · Reviewer_W2qU · 2023-11-21
> >
> > Thanks for the responses. My concerns have been addressed. Therefore, I raised my rating.

---

> > > ### Author Response · Authors · 2023-11-21
> > > **Thank you for your response**
> > >
> > > Thank you for kindly giving us feedback on our author's response.
> > > We are glad to hear that the response clarified the concerns.
> > >
> > > We would like to thank you again for your help in improving the manuscript.

---

### Official Review · Reviewer_C5D8 · 2023-11-05

**Soundness:** 3 good
**Presentation:** 3 good
**Contribution:** 3 good
**Rating:** 6
**Confidence:** 3

**Summary:**

This paper introduces Structured Activation Sparsification (SAS), a method that enhances the accuracy of wide neural networks without the additional computational cost typically associated with network width. By implementing structured sparsity within activation maps—where a set number of non-zero values are maintained in consecutive activation. This allows for the simplification of wide matrix multiplications into narrow ones. Empirical results show that this method can improve accuracy (by up to 7% on CIFAR10) without increasing computational demands and outperforms similar sparsity approaches applied to network weights.

**Strengths:**

The strengths of this work is listed below:
1. This work introduces a novel method of structuring sparsity in activations, which appears to enable a reduction in computation without a corresponding drop in accuracy. The concept of using structured sparse projection (SAS) to maintain vectorization compatibility is particularly innovative.
2. The structured sparsity allows for efficient matrix multiplication operations that maintain the number of multiplications at the level of the base dense/narrow network, highlighting efficiency in computation.
3. The process for creating sparse activations through the structured sparse projection S is described as having negligible computational cost and wall-clock latency, indicating an efficient method that does not add significant complexity or processing time.
4. A thorough evaluation is presented that demonstrates the SAS network's increased expressiveness compared to Static Weight Sparsity (SWS) networks, given the same computational budget, by using trajectory length analysis.
5. The proposed SAS projection and its integration into the training process suggest a straightforward transformation of existing neural networks to increase their efficiency, which could be widely applicable across different network architectures and tasks.

**Weaknesses:**

The weakness of this work is listed below:
1. As mentioned in Section 2.3, while the computational load in terms of FLOPS remains the same for a given level of activation sparsity
M, the memory requirements for the Sparse Activation Sparsification (SAS) network increase linearly with M. This is in contrast to the Sparse Weight Storage (SWS) network, which maintains a constant storage requirement for weights at inference time regardless of M. This could become a significant limitation for devices with limited memory or when scaling to very large networks.
2. In Section 3, the paper discusses the expressive power of SAS networks by comparing trajectory lengths in a specific constructed neural network with 2-dimensional input and output. However, this analysis might not generalize to all network architectures or datasets, which could limit the understanding of the practical implications of SAS's increased expressive power.
3. The straightforward method used for computing the index I might not be the most effective approach, particularly when neighbor elements oscillate around zero (Section 6.1). An indexing strategy that is not learned end-to-end may limit the model's capacity to adapt to the data's complexity, potentially leaving some performance on the table.
4. In general, the listings (code pieces) in the paper is informative enough. However, I would suggest the authors to replace the source code with some high-level pseudo code. This is more readable and more accessible to some readers.

**Questions:**

I do not have other questions. Please refer to the weakness column for my comments.

---

> ### Author Response · Authors · 2023-11-17
> **Response (1/1)**
>
> We appreciate the positive and encouraging feedback and insightful comments from *C5D8*.
> We answer each of the valuable comments as follows.
>
> >As mentioned in section 2.3, while the computational load in terms of FLOPS remains the same for a given level of activation sparsity $M$, the memory requirements for the Structured Activation Sparsification (SAS) network increase linearly with $M$.
> This is in contrast to the Structured Weight Sparsification (SWS) network, which maintains a constant storage requirement for weights at inference time regardless of $M$.
> This could become a significant limitation for devices with limited memory or when scaling to very large networks.
>
> As *C5D8* points out, in contrast to SWS, the memory requirement of SAS increases linearly with $M$ (section 2.3).
> Actually, this is the main drawback of the proposed method  (section 2.3, section 4.2).
> We expect this could be mitigated by using quantized weight (section 5), which we left for future work.
> From a practical perspective, commercially available hardware (at the time of submission) supports 50\% sparsity; doubling the memory might be a practical choice, especially when the FLOPS is critical, considering the significant accuracy gain from M=0 (dense) to $M$=2 (50\% sparse).
> We clarified this point in (section 2.3 and section 5).
>
>
> >In section 3, the paper discusses the expressive power of SAS networks by comparing trajectory lengths in a specific constructed neural network with 2-dimensional input and output. However, this analysis might not generalize to all network architectures or datasets, which could limit the understanding of the practical implications of SAS's increased expressive power.
>
> We agree with *C5D8* about the interpretation of the trajectory length analysis.
> Still, we consider the analysis gives important implications about the expressive power of the model, which does not depend on data, model, or several training details that affect the accuracy (It often happens, for example, model A performs better than B when using a specific optimizer, learning rate, and learning epochs; but model B beats model A when adopting different training scheme).
> The trajectory length analysis does not depend on these settings.
> In section 4, we present results using a practical model and dataset, and we hypothesize the increased accuracy by adopting the SAS is attributed to the increased expressive power, which conforms to the trajectory length analysis results.
> Thanks to the *C5D8*'s comments, we find the above motivation for utilizing the trajectory length analysis was missing in the original manuscript (we only mentioned "more generic settings"); we've added the above discussion at the beginning of the trajectory length analysis (section 3).
>
> >The straightforward method used for computing the index I might not be the most effective approach, particularly when neighbor elements oscillate around zero (section 6.1). An indexing strategy that is not learned end-to-end may limit the model's capacity to adapt to the data's complexity, potentially leaving some performance on the table.
>
> We agree on this point; however, the oscillation phenomenon happens when $M$>2 and will be several when it is large, e.g., >16 (section 6.1).
> We consider the proposed simple index computation mechanism of e.q (3) to be one of the most effective approaches, at least for $M$=2, because there are no oscillations, and it is computationally efficient.
> Thanks to the  *C5D8*’s comment, we noticed that the original manuscript could be more precise on this point.
> We've added the above discussion in section 6.1.
>
> >In general, the listings (code pieces) in the paper are informative enough. However, I would suggest the authors replace the source code with some high-level pseudo code. This is more readable and more accessible to some readers.
>
> We present the source code for inference to demonstrate that the SAS operation is realizable on actual commercial hardware (listing-1).
> We also show the source code for training to demonstrate the conversion from the existing dense layer to the SAS layer for training is simple and trivial (listing-2).
> We agree with *C5D8*'s suggestion for the importance of high-level explanation.
> Following the suggestion, we further clarified the method during the training, and now we consider listing-2 to be somewhat redundant; therefore, we moved it to supplement for interested readers (and added more important discussion regarding the valuable comments from all the reviewers).
> Regarding listing-1 (Code for inference on actual hardware), we keep it in the main text to showcase that the SAS could actually be compatible with commercial hardware.
> However, thanks to the *C5D8*'s comments, we noticed some details could be more precise to follow; therefore, we've clarified the high-level explanation in sections 2.1-2.3 and supp. E along with the visual description.

---

> > ### Comment · Reviewer_C5D8 · 2023-11-20
> > **Thank you for your response and clarification.**
> >
> > I have read through the rebuttal and I do not have other questions. I am positive about this work but I will not further increase my score. Thanks the authors again.

---

> > > ### Author Response · Authors · 2023-11-21
> > > **Thank you for your response**
> > >
> > > Thank you for your kind feedback and encouraging comments.
> > > We are glad to hear that the response clarified the concerns.

---

### Author Response · Authors · 2023-11-17
**Thank you for the insightful feedback**

We thank all the reviews for their insightful feedback.
We appreciate that all the reviews agree on the novelty of the proposed concept of SAS, the utilization of structured sparsity in activation.
We answered each valuable comment in each thread to clarify the reviewer's concerns.

We are now sure that the revised manuscript incorporates all the feedback.
Please let us know if there are still unclear discussions.
Also, we are happy to answer any additional questions.

Thank you again for the valuable feedback to make the paper more clear and strong.

---

### Meta-Review · Area_Chair_FN8M · 2023-12-12

**Metareview:**

This paper introduces Structured Activation Sparsification (SAS), an innovative method to improve model accuracy without increasing computational costs. The core idea is to implement structured sparsity within activation maps, allowing simplification of wide matrix multiplications. Overall, reviewers acknowledge the novelty of this approach and its potential for efficiency gains. The concept of structured sparse projection to maintain vectorization compatibility is considered technically sound. Another strength noted is the straightforward integration of SAS into existing architectures.

However, concerns are raised regarding increased memory requirements and insufficient evidence generalizing SAS's benefits. One reviewer rightfully points out ambiguities around computational equivalence claims between SAS and static weight sparsity techniques. Some open questions also remain about the method's memory overhead and general applicability.

Considering the reviewers' scoring and rebuttal discussions, I would lean toward acceptance given the conceptual innovation presented. However, addressing the identified weaknesses through additional experiments, analyses, and clarified explanations is critical. Moreover, the authors should temper the claims around computational equivalence until fully substantiated.

**Justification For Why Not Higher Score:**

All reviewers give lukewarm scores. Concerns are raised regarding increased memory requirements and insufficient evidence generalizing SAS's benefits. One reviewer rightfully points out ambiguities around computational equivalence claims between SAS and static weight sparsity techniques. Some open questions also remain about the method's memory overhead and general applicability.

**Justification For Why Not Lower Score:**

I would recommend accepting this paper with major revisions focused on shoring up the identified issues.

---

### Decision · Program_Chairs · 2024-01-16

Accept (poster)